# Patient Safety Culture Assessment in Primary Care Settings in Greece

**DOI:** 10.3390/healthcare9070880

**Published:** 2021-07-13

**Authors:** Ioannis Antonakos, Kyriakos Souliotis, Theodora Psaltopoulou, Yannis Tountas, Maria Kantzanou

**Affiliations:** 1Laboratory of Hygiene, Epidemiology and Medical Statistics, University of Athens Medical School, 115 27 Athens, Greece; 2Faculty of Social & Political Sciences, University of Peloponnese, 20131 Corinth, Greece; ksouliotis@uop.gr; 3Department of Hegiene, School of Medicine, University of Athens, 115 27 Athens, Greece; tpsaltop@med.uoa.gr (T.P.); ytountas@med.uoa.gr (Y.T.); maria.kantzanou@gmail.com (M.K.)

**Keywords:** primary care, patient safety culture, Agency of Healthcare Research and Quality

## Abstract

Introduction: A positive safety culture is considered a pillar of safety in health organizations and the first crucial step for quality health services. In this context, the aim of this study was to set a reference evaluation for the patient safety culture in the primary health sector in Greece, based on health professionals’ perceptions. Methods: We used a cross-sectional survey with a 62% response rate (*n* = 459), conducted in primary care settings in Greece (February to May 2020). We utilized the “Medical Office Survey on Patient Safety Culture” survey tool from the Agency for Healthcare Research and Quality (AHRQ). The study participants were health professionals who interacted with patients from 12 primary care settings in Greece. Results: The most highly ranked domains were: “Teamwork” (82%), “Patient Care Tracking/Follow-up” (80% of positive scores), and “Organizational Learning” (80%); meanwhile, the lowest-ranked ones were: “Leadership Support for Patient Safety” (62%) and “Work Pressure and Pace” (46%). The other domains, such as “Overall Perceptions of Patient Safety and Quality” (77%), “Staff Training“ (70%), “Communication about Error” (70%), “Office Processes and Standardization” (67%), and “Communication Openness” (64%), ranked somewhere in between. Conclusions: A positive safety culture was identified in primary care settings in Greece, although weak areas concerning the safety culture should be addressed in order to improve patient safety.

## 1. Introduction

Hippocrates’ famous statement “To help or, at least, not to harm” 25 centuries ago indicates that concern for the safety of patients is not a recent phenomenon. Since its origins, “to not cause harm” has been a fundamental principle of medicine, although it implicitly recognizes the possibility that its actions produce undesirable consequences. Despite the accessibility and continuity of assistance in primary care (PC), studies performed at this level are scarcer than in middle hospitals [1,2]. However, the majority of medical consultations take place in primary care, and many of the adverse events in hospitals may be initiated there, making the need for research into primary care patient safety even more significant. In order to identify and classify patient safety issues in primary care, there is no single standard [3]. The literature indicates that 24–85% of all key adverse events are preventable [4]. Moreover, evidence shows that 50% of primary care harm is avoidable in high-income countries, while 60% is avoidable in low-income countries [4]. Achieving a culture of safety is a crucial first step. This involves an appreciation of the principles, expectations, and standards of what is important in institutions and what actions and activities are associated with patient safety [5]. Communication centered on trust and respect, common views of the significance of safety, and belief in the success of prevention methods distinguish organizations with a positive safety culture [5]. In 2012, the WHO noted that patient safety in primary care is a problem that requires local and long-term solutions, and that one of the first steps should be to assess the patient safety culture [6].

Measuring a safety culture, especially from health professionals’ point of view, enables the identification of strengths and areas for improvement. It also enables the development of appropriate interventions to evaluate new safety programs by comparing results before and after implementation [7,8,9].

Primary care is considered the point of entrance into the Greek healthcare system. The care delivered in Greek primary care settings has a direct impact on Greek families’ wellbeing and the utilization of financial infrastructure. As a result, unstable, inadequate, or incompetent primary care can lead to unnecessarily high morbidity and mortality, as well as a waste of hospital resources [10,11].

Although a few studies have been conducted in Greece that assess patient safety culture in hospital settings [12,13,14], no study has been carried out in a primary care setting. In this study, we examined healthcare professionals’ answers to a validated research tool dedicated to this task: The Medical Office on Patient Survey Culture (MSOPSC), of the Agency of Health Research and Quality (AHRQ) [15]. The MSOPSC was selected as the tool for testing instead of various others, such as the Hospital Survey on Patient Safety Culture (HSOPSC) [16], the Manchester Patient Safety Framework (MaPSaF) [17], or the Safety Attitudes Questionnaire (SAQ) [18], because it is validated and dedicated to primary care settings. The slight differences in safety culture aspects among these tools [16,17,18] are shown in Table 1. In addition to the questionnaire, AHRQ provides a 2020 database comprising data from 18.396 respondents from 1.475 Medical Offices in the U.S. that use the MSOPSC survey [19].

The purpose of this study was to analyze safety culture scores for primary health facilities in Greece in order to set a baseline assessment of the patient safety culture and to identify opportunities for improvement. A comparison of the findings with the current benchmark score data was also performed.

## 2. Materials and Methods

### 2.1. MSOPSC Measurement Tool

This questionnaire was constructed as a self-assessment tool for health practitioners and features issues concerning their perceptions of patient safety and quality of care. It has been validated and utilized by researchers all around the world [20,21]. It contains 38 items measuring 10 aspects of the culture of patient safety, nine items relating to “Patient Safety and Quality Issues”, four items relating to “Information Exchange with Other Settings”, and four items relating to “Average Overall Ratings on Quality and Patient Safety”.

The total percentage of respondents within a PHU who gave one of the following answers on items with five-point response scales is recognized as a percent positive response: “Strongly agree”/“Agree” or “Always”/“Most of the time”. Because a negative response on a negatively phrased item represents a positive response, the corresponding proportion for negatively worded questions is the total percentage of respondents within a PHU who answered “Strongly disagree”/”Disagree” or “Never”/”Rarely”.

The “Patient Safety and Quality Issues” and “Information Exchange with Other Settings” components’ percent positive values were computed differently than the other survey questions. The total of the three response options that represent the smallest frequency of occurrence determines the percent positive score for these 13 items. Scores of more than 75% are regarded as positive indicators of safety.

The questionnaire was translated from English to Greek, checked by professionals, and tested in a pilot study that revealed that all dimensions and items should be retained, except for the dimension “Information Exchange with Other Settings”, which was deleted due to a high non-response rate and non-applicability.

### 2.2. Study Design

Random stratified sampling was conducted among the primary care units of the first health region of Attica, Greece. Twelve out of 78 PHU’s were selected in a representative way in terms of services provided and health professionals’ specialties. Participants were random selected among the twelve health facilities.

The eligibility criteria for the respondents were to be a multidisciplinary team professional, who assisted the patient with direct and indirect assistance, had served in the unit for at least 30 days, and worked at least 20 h per week. Respondents who did not meet the requirements set out above were exempt from the study. During working hours, the survey was given to the chosen participants in a sealed envelope and gathered in a sealed envelope two to three days after delivery. To prevent peer impact, the participants were told not to share the questionnaire with one another. As a member, the intermediary was not included.

Originally, the MSOPSC was developed to refer to all professionals [15]. The pre-test, however, found that workers who were not directly engaged in patient care sometimes did not respond to things that dealt with actual patient care (i.e., managers and administrators). Consequently, 469 participants returned the survey (response rate = 60%), and 10 questionnaires were omitted, in which less than half of the items were answered. Finally, for further study, 459 questionnaires were maintained (Table 2).

### 2.3. Statistical Analysis

IBM SPSS 21.0 software (IBM Corp., 2012) was used to conduct the statistical analysis (IBM SPSS for Windows, Version 21.0; IBM Corp., Armonk, NY, USA). Absolute (*N*) and relative (%) frequencies were used in the descriptive analysis of the categorical variables.

Cronbach’s alpha coefficient was used to assess the MSOPSC’s reliability (Table 3). Cronbach’s alpha is recognized as one of the most essential and prevalent statistics in test construction and research [22], to the point that its usage in multiple-item measures research is considered commonplace [23,24].

The PHU items and composite scores are shown in Table 3 and Figure 1. The overall ratings of the quality and safety assessment are shown in Table 4. All of the results are compared with results from medical offices in the U.S.

A statistically significant correlation between categorical variables is defined as a *p*-value of ≤0.05.

## 3. Results

Female professionals (58.8%) aged between 31 and 50 years (74.7%), married or living with a partner (64%), and with a Bachelor’s degree (75%) predominated. As for the professional category, 230 (50.1%) were physicians, 85 (18.5%) were nurses, and 144 (31.44%) were supporting clinical staff (radiographer, health supervisor, and nurse assistant). Regarding work experience and the work area of the respondents, 50.8% had more than 10 years of experience, 35% worked in medical departments (family practice, pathology, and pediatrics), 33% worked in microbiology, and 15% worked in radiology departments (Table 2).

The most highly ranked composites by the respondents were “Teamwork” (82% positive rating), “Patient Care Tracking/Follow-up” (80%), “Organizational Learning” (80%), and “Overall Perception of Patient Safety and Quality” (78%). “Staff training” (70% of positive responses), “Communication About Errors” (70%), “Office Processes and Standardization” (67%), and “Communication Openness” (64%) followed. The lowest scores were for “Owner/Managing Partner/Leadership Support for Patient Safety” (62%) and “Work Pressure and Pace” (46%) (Figure 1 and Table 3).

The overall ratings of quality and patient safety received positive scores, with 73% and 70% of participants returning “very good” or “excellent” responses, respectively (Table 4).

## 4. Discussion

To the best of our knowledge, this is the first study to evaluate the safety culture in primary care settings in Greece by utilizing the MSOPSC tool, which is dedicated to primary healthcare.

The safety culture domains that received the highest rankings in this survey were: “Teamwork” (82%), “Patient Care Tracking/Follow-up” (80%), and “Organizational Learning” (80%). The operation of primary healthcare units in Greece in small multidisciplinary teams seems to strengthen the spirit of cooperation between employees. This is not only reflected in the high percentage associated with the “Teamwork” dimension, but also in the “Organizational Learning” dimension. This is because interdisciplinary collaboration helps health professionals understand shared contextual roles and responsibilities, and thus execute organizational goals, interact with and distribute pertinent information, and provide safe and effective care. “Teamwork” emerged as the highest safety culture domain in Yemen (96%) [25] and in Holland (79.2%) [26], where the MSOPSC and SAQ questionnaires were used, respectively.

Another strong area in the current study was the “Patient Care Tracking/Follow-up”. This indicates that PHU patients in Greece are reminded of their dates, their adherence to the therapeutic process is verified, and the follow-up with patients who require monitoring is adequate. This is largely due to the fact that electronic systems in primary care have been modernized in recent years (to include patients’ electronic files and electronic prescriptions); however, a lot of work is still necessary, especially in the interconnection of the primary and secondary health sectors. Similar results were reported in countries with modern health information systems, such as the U.S. (88%) [19] and Spain (77%) [27], while lower results were reported in countries such as Yemen (52%) [25] and Poland (65%) [28], where primary care services are not supported by an information system.

In contrast, the least positive responses were for the “Work Pressure and Pace” (46%) and “Leadership Support” (62%) domains. This was mainly due to the understaffing of nurses, a chronic problem in the healthcare system in Greece. According to the WHO [29], in Greece there are 3.6 nurses per 1000 population, compared to 9.1 nurses per 1000 in the OECD. Switzerland, Norway, and Denmark all have more than 16 nurses per 1000 residents, with Switzerland demonstrating the highest ratio with 17.4 nurses per 1000 population. The lack of leadership support highlights the need to bridge the communication gap between the management and employees. Effective leadership that promotes the safety culture must combine the basic attributes of effective communication, collaboration, experience, and adaptability [30]. It is noteworthy that in similar studies, countries with very different cultures have the highest rates in this domain, such as Iran [31] (75%), Holland [26] (73%), and Poland [28] (84%).

The majority of participants said their training at basic healthcare facilities was beneficial. When asked about the availability of on-the-job training, the American respondents fared worse in terms of new procedures and being asked to conduct activities for which they were not qualified; however, they fared better when asked about the availability of on-the-job training [32]. In this sector, the current study seemed to have an average of 70%, which was much higher than Spain [27] and Yemen [25] with 61%, but lower than Poland [28] with 79%.

This study found no significant relationship between any of the safety culture composites and gender. In contrast, a Polish study [28] revealed that women rated domains such as “Patient Care Tracking/Follow-up” and “Overall Perception of Patient Safety and Quality” slightly higher than men. Men had better outcomes in the “Information Exchange with Other Settings” domain. Women in Spain scored higher than men in the following areas: “Information Exchange with Other Settings”, “Work Pressure and Pace”, “Staff Training”, “Office Processes and Standardization”, “Patient Care Tracking/Follow-up”, “Leadership Support for Patient Safety”, and “Overall Perception of Patient Safety”. Concerning patient safety, men had slightly better overall rating scores. Opposite findings were observed in Iranian participants, whereas women’s perceptions regarding patient safety were substantially higher.

The current study found that respondents with more than 10 years of experience at their medical office provided a better assessment of the following aspects of patient safety dimensions: “Owner/Managing Partner/Leadership Support for Patient Safety” and “Communication Openness”. The American study found no association between seniority and responses relating to patient safety [32]. The survey respondents valued the overall quality at their medical office better if they had less than 10 years of experience. The opposite association was observed in Spain, where 10+ years of experience resulted in worse patient safety scores from respondents [33]. A positive correlation between age and the domains of “Teamwork”, “Patient Safety and Quality”, and “Communication Openness” was found in a Dutch study [20].

Future research should explore the causes of the disparities between healthcare providers. Moreover, it should establish whether there is a link between the patient safety culture, patient experiences, the overall rating on safety and quality, and the occurrence of adverse events in general practice cooperatives. In addition, investigations examining the impacts of interventions aimed at improving communication or creating a safer environment are advised.

Overall, the ratings for healthcare safety and quality in this study were satisfactory in all areas (more than 70%) except for timeliness and efficiency; these were positively rated by 65% and 63% of respondents, respectively (Table 3).

In summary, the positive ratings for the patient safety culture at primary care settings in Greece were mostly reflected in the domains “Teamwork”, “Patient Care Tracking/Follow-up”, and “Overall Perceptions of Patient Safety”. Thereafter, the domains “Staff Training”, “Communication About Error”, “Office Processes and Standardization”, and “Communication Openness” followed. Respondents provided poor ratings for “Leadership Support for Patient Safety” and “Work Pressure and Pace”. Specific areas related to the patient safety culture received better scores from younger participants and those who had less than 10 years of experience.

This study has several strengths and potential limitations that should be considered. One of the strengths is the use of the MSOPSC, the most commonly utilized instrument for evaluating the safety culture in primary care settings. However, investigating the relationship between the patient safety composite scores and the respondents’ features with the safety culture outcomes (in Greece) is not common in the literature. To the best of our knowledge, this is the first study in Greece to explore this scenario. However, a range of constraints on these data should be considered. First, the units that provided the data were not a random sample of all PHUs in the country, but a stratified random sampling of units from the first health region in Greece: Attica. Therefore, a positive selection bias is possible. Second, the results revealed in this study represent the views of health professionals in primary care settings and not administrative and technical staff. Finally, no effort was made to assess the validity of the evidence provided by the units against other evaluation reports, e.g., interviews or record reviews [34].

## 5. Conclusions

This is the first study conducted in primary care settings in Greece to evaluate the patient safety culture. Overall, a positive safety culture was identified among health professionals, although some weak areas should be addressed in order to improve patient safety. Further studies should investigate the differences between professional subgroups in order to provide interventions and to improve safety issues.

## Figures and Tables

**Figure 1 healthcare-09-00880-f001:**
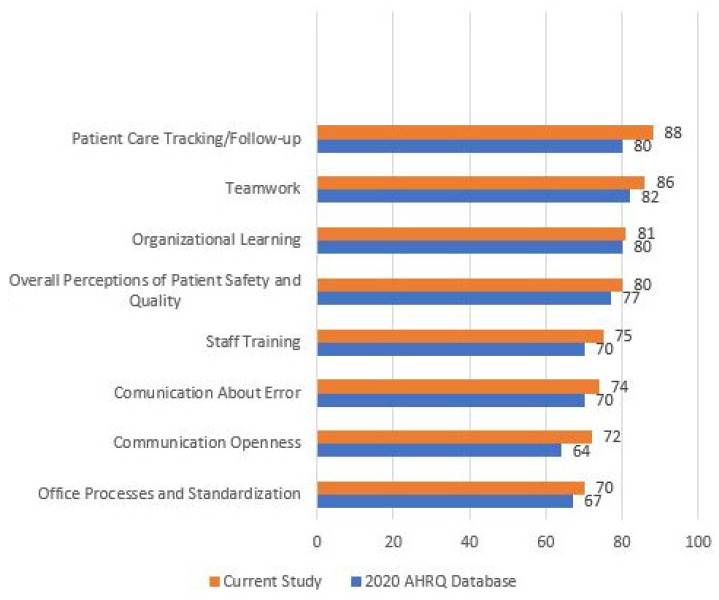
Comparison of patient safety culture scores for Greek primary care centers with U.S Medical Offices.

**Table 1 healthcare-09-00880-t001:** Comparison of the safety culture dimensions measured by different survey tools.

MSOPSC *	HSOPSC **	MaPSaF ***	SAQ ****
Teamwork	Frequency of error reporting	Continuous improvement	Job satisfaction
2. Patient care tracking/follow-up	2. Number of reported errors	2. Priority given to staff	2. Safety climate
3. Organizational learning	3. Supervisors’ expectations and actions	3. System errors and individual responsibility	3. Teamwork climate
4. Overall perceptions of patient safety and Quality	4. Organizational learning	4. Recording incidents	4. Working conditions
5. Staff training	5. Teamwork within units	5. Evaluation incidents	5. Preparation of management
6. Owner/managing partner/leadership support for patient safety	6. Communication openness	6. Learning and effecting change	6. Stress recognition
7. Communication about error	7. Feedback and communication about errors	7. Communication personnel management	
8. Communication openness	8. Non-punitive response to errors	8. Staff education	
9. Office processes and standardization	9. Staffing	9. Teamwork	
10. Work pressure and pace	10. Management support		
	11. Teamwork across units		
	12. Handoffs and transitions		

* Developed by the U.S. Agency for Healthcare and Research [15]. ** Developed by the U.S. Agency for Healthcare and Research [16]. *** Developed by the University of Manchester [17]. **** Developed by the University of Texas [18].

**Table 2 healthcare-09-00880-t002:** Respondents’ characteristics.

Participants’ Characteristics	*N* (%)
Males	189 (41.2)
Females	270 (58.8)
Age (years)	*N* (%)
31–40	153 (33.3)
41–50	184 (41.4)
51–60	107 (23.3)
61–70	15 (3.3)
Marital status	*N* (%)
Married or with a partner	344 (75%)
Not married or without a partner	115 (25%)
Work experience (years)	*N* (%)
<10	226 (49.2)
≥10	233 (50.8)
Staff position	*N* (%)
Physician	230 (50.1)
Nurse	85 (18.5)
Supporting clinical staff *	144 (31.4)
Work area/unit	*N* (%)
Medical services **	35 (52)
Microbiology	22 (33)
Radiology	10 (15)

* Radiographer, health supervisor, and nurse assistant; ** family practice, pathology, and pediatrics.

**Table 3 healthcare-09-00880-t003:** Item-level and Cronbach’s alpha results for Greek primary care centers (*N* = 78) and U.S. medical offices (*N* = 1.475).

Item	Survey Items by Patient Safety Culture Area	Current Study	2020 AHRQ Database
	1. Teamwork	Cronbach’s alpha = 0.82	Cronbach’s alpha = 0.83
C1	When someone in this office gets really busy, others help out.	78	86
C2	In this office, there is a good working relationship between staff and providers.	84	90
C5	In this office, we treat each other with respect.	83	85
C13	This office emphasizes teamwork in taking care of patients.	82	85
	2. Patient Care Tracking/Follow-up	Cronbach’s alpha = 0.74	Cronbach’s alpha = 0.78
D3	This office reminds patients when they need to schedule an appointment for preventive or routine care.	86	88
D5	This office documents how well our chronic-care patients follow their treatment plans.	73	80
D6	Our office follows up when we do not receive a report we are expecting from an outside provider.	76	86
D9	This office follows-up with patients who need monitoring.	86	91
	3. Organizational Learning	Cronbach’s alpha = 0.80	Cronbach’s alpha = 0.82
F1	When there is a problem in our office, we see if we need to change the way we do things.	76	83
F5	This office is good at changing office processes to make sure the same problems do not happen again.	83	79
F7	After this office makes changes to improve the patient care process, we check to see if the changes worked.	80	74
	4. Overall Perceptions of Patient Safety and Quality	Cronbach’s alpha =0.70	Cronbach’s alpha = 0.74
F2	Our office processes are good at preventing mistakes that could affect patients.	81	85
F3R	Mistakes happen more than they should in this office.	77	77
F4R	It is just by chance that we do not make more mistakes that affect our patients.	76	77
F6R	In this office, getting more work done is more important than quality of care.	75	70
	5. Staff Training	Cronbach’s alpha = 0.72	Cronbach’s alpha = 0.63
C4	This office trains staff when new processes are put into place.	69	76
C7	This office makes sure staff get the on-the-job training they need.	70	75
C10R	Staff in this office are asked to do tasks they have not been trained to do.	71	66
	6. Owner/Managing Partner/Leadership Support for Patient Safety	Cronbach’s alpha = 0.71	Cronbach’s alpha = 0.76
E1R	They are not investing enough resources to improve the quality of care in this office.	61	63
E2R	They overlook patient care mistakes that happen over and over.	64	64
E3	They place a high priority on improving patient care processes.	79	82
E4R	They make decisions too often based on what is best for the office rather than what is best for patients.	77	79
	7. Communication About Error	Cronbach’s alpha = 0.72	Cronbach’s alpha = 0.80
D7R	Staff feel like their mistakes are held against them.	68	73
D8	Providers and staff talk openly about office problems.	63	73
D11	In this office, we discuss ways to prevent errors from happening again.	63	73
D12	Staff are willing to report mistakes they observe in this office.	63	59
	8. Communication Openness	Cronbach’s alpha = 0.72	Cronbach’s alpha = 0.80
D1	Providers in this office are open to staff ideas about how to improve office processes.	64	61
D2	Staff are encouraged to express alternative viewpoints in this office.	79	68
D4R	Staff are afraid to ask questions when something does not seem right.	53	56
D10R	It is difficult to voice disagreement in this office.	82	84
	9. Office Processes and Standardization	Cronbach’s alpha = 0.71	Cronbach’s alpha = 0.78
C8R	This office is more disorganized than it should be.	51	47
C9	We have good procedures for checking that work in this office was done correctly.	67	78
C12R	We have problems with workflow in this office.	75	80
C15	Staff in this office follow standardized processes to get tasks done.	54	59
	10. Work Pressure and Pace	Cronbach’s alpha = 0.72	Cronbach’s alpha = 0.76
C3R	In this office, we often feel rushed when taking care of patients.	27	38
C6R	We have too many patients for the number of providers in this office.	34	45
C11	We have enough staff to handle our patient load.	56	46
C14R	This office has too many patients to be able to handle everything effectively.	68	68

**Table 4 healthcare-09-00880-t004:** Overall rating of quality and safety in the current study (*n* = 459) as compared with the 2020 Medical Office Database (*n* = 18.396).

		This Study (%)	AHRQ, 2020 Database (%)
		Excellent/Very Good	Excellent/Very Good
Overall rating of quality issues	Patient-centered	75	71
	Effective	75	71
	Timely	65	56
	Efficient	63	62
	Equitable	85	84
Average rating of quality issues		73	69
Overall rating of patient safety		70	68

## Data Availability

Not applicable.

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
