# Peer review of "Patient Safety Culture Assessment in Primary Care Settings in Greece"

_healthcare, 2021, doi:10.3390/healthcare9070880_

Round 1
Reviewer 1 Report
I have carefully reread the manuscript I greatly appreciate the efforts of the authors in improving the manuscript. Actually, with the corrections and improvements made, I think it is suitable to be accepted for publication.
Author Response
We thank the reviewer for his kind comments and his positive proposal for the publication of the study.
Reviewer 2 Report
Overall, this paper has received a major rewrite and is substantially improved. The study adapts a patient safety culture survey developed by the US Agency for Healthcare Research and Quality for use in Greece. The authors explain the rationale for selection of this instrument. It does appear to be appropriate and has been validated in the US.
The abstract and the paper refer to the application of Cronbach’s alpha to measure reliability. As previously noted, Dr. Cronbach preferred to call the measure “coefficient alpha” or “raw coefficient alpha.” More recent studies have observed that the measure should not be used unconditionally. Other reliability coefficients based on structural equation modeling are recommended as an alternative. That said, coefficient offer remains one of the most commonly used reliability coefficients and the technique in this paper is appropriate.
The authors use Pearson correlation to compare relationships between survey items and overall patient safety. The assumptions for use of Pearson include interval or ratio of data, normal distributions, linear relationships, and the absence of outliers. The authors should state how they determined that the data used for this study meet the assumptions required for use of Pearson. Otherwise, they might want to conduct a Spearman correlation rather than a Pearson correlation.
In the study design section on page 3 I would like to see some discussion of how participants were selected geographically. I also do not understand the sentence “The chosen practitioners were explained – by an intermediary, the head nurse, and the principal investigator – the importance of the survey and its possible effect on improvement of primary healthcare.”
I’m not sure that displaying reliability coefficients for each item of the instrument is very effective. I think that the authors could just describe these results in a paragraph rather than taking up three pages to display them numerically. The overall rating of quality issues on page 8 is the information that is most pertinent to the reader. Most of the individual questions appear to be reliable with the exception of C3R, C6R and C11 where the coefficients are pretty weak. The authors might note that the findings that include problems with respondents feeling is based on responses to questions that may not be particularly reliable.
Table 5 on page 8 is the most problematic for me. This is a Pearson correlation to consider relationships between subscales of the instrument and overall rating of safety. They question about the use of Pearson comes up here. These correlations are pretty weak the exception of teamwork which might be somewhere between moderate and weak with a coefficient of 0.40. Also, I assume that these columns are correlations for Greece and for the US but they are not labeled as such. Interestingly, the correlation coefficient for Greece for teamwork is substantially higher than for the US. Also, the US correlations appear to be generally weak as well.
The authors professionally point out the problem that the study participants were not randomly selected. The method of selection should be set forth in the study design section.
The rewrite itself has very much improved the paper. This paper would contribute to the literature in terms of providing a patient safety survey in a primary care setting and in terms of providing a patient safety perception survey among health professionals in Greece.
Author Response
We appreciate the time and effort you dedicated to providing feedback on our manuscript and are grateful for the insightful comments on and valuable improvements to our paper. We have tried to incorporate your suggestions. Those changes are highlighted within the manuscript. Please see below, in red, for a point-by-point response to your comments and concerns. Page numbers and lines refer to the revised manuscript.
- Comment from Reviewer #2:
“The authors use Pearson correlation to compare relationships between survey items and overall patient safety. The assumptions for use of Pearson include interval or ratio of data, normal distributions, linear relationships, and the absence of outliers. The authors should state how they determined that the data used for this study meet the assumptions required for use of Pearson. Otherwise, they might want to conduct a Spearman correlation rather than a Pearson correlation..”
Author response:
Thanks to the reviewer for this apposite remark. Indeed, the data of the study are suitable for Spearman correlation instead of Pearson for the reasons that the reviewer mentioned. We conducted a Spearman correlation in order to investigate any significant relationship between subscales of the instrument and overall rating of safety, but no significant relationship revealed.
We agree with the reviewer that Table 5 is problematic and causes confusion to the reader. So, we deleted Table 5 and lines 165-172 that refer to Table 5.
Also, we modified the sentence in line 50
2. Comment from Reviewer #2:
“In the study design section on page 3 I would like to see some discussion of how participants were selected geographically. I also do not understand the sentence “The chosen practitioners were explained – by an intermediary, the head nurse, and the principal investigator – the importance of the survey and its possible effect on improvement of primary healthcare.”
Author response:
Thanks to the reviewer for this remark. We have addressed this issue by adding a small paragraph (lines 105-108)
Concerning the sentence mentioned by the reviewer, we decided to be deleted in order to avoid any confusion to the reader.
3. Comment from Reviewer #2:
“I’m not sure that displaying reliability coefficients for each item of the instrument is very effective. I think that the authors could just describe these results in a paragraph rather than taking up three pages to display them numerically. The overall rating of quality issues on page 8 is the information that is most pertinent to the reader. Most of the individual questions appear to be reliable with the exception of C3R, C6R, and C11 where the coefficients are pretty weak. The authors might note that the findings that include problems with respondents' feelings are based on responses to questions that may not be particularly reliable. “
Author response:
We understand the reviewer’s concerns about the extent of Table 3. But this table as well as Figure 1 present a complete picture of the results of the study. In addition, AHRQ data (see reference [16]), which is the biggest database of this kind of study presents the results in this structure. So, for benchmark purposes too, we think that Table 3 should retain.
Regarding the individual questions C3R, C6R, and C11:
Indeed, questions C3R and C6R (not C11) appear low rating scores (positive responses from the participants) compared to AHRQ data. This may be due to Cronbach’s alpha (0.72 is an acceptable value but not an excellent one).
- Comment from Reviewer #2:
“Table 5 on page 8 is the most problematic for me. This is a Pearson correlation to consider relationships between subscales of the instrument and overall rating of safety. They question about the use of Pearson comes up here. These correlations are pretty weak the exception of teamwork which might be somewhere between moderate and weak with a coefficient of 0.40. Also, I assume that these columns are correlations for Greece and for the U.S but they are not labeled as such. Interestingly, the correlation coefficient for Greece for teamwork is substantially higher than for the US. Also, the US correlations appear to be generally weak as well. “
Author response:
As we mentioned in comment #1 we deleted table 5, because it doesn’t contribute significantly to the study.
5. Comment from Reviewer #2:
The authors professionally point out the problem that the study participants were not randomly selected. The method of selection should be set forth in the study design section.”
Author response:
As we mentioned in comment #2 have addressed this issue at lines 105-108 in the study design section.

This manuscript is a resubmission of an earlier submission. The following is a list of the peer review reports and author responses from that submission.
Round 1
Reviewer 1 Report
First, the title "Patient safety culture assessment in primary care settings in Greece during the pandemic of COVID -19" does not correspond to the reality of the manuscript. The Covid19 context is named in the title, in the abstract and in the conclusion, but at no time, in reading the manuscript, does it refer to the investigation having anything to do with the pandemic. In fact, the project was approved by the corresponding ethics committee in 2019, before the pandemic.
the aim of this study was to evaluate safety culture scores for primary health facilities, but finally the conclusions speak of the adaptation and validation of the MSOPSC questionnaire into Greek. Aspect that must be clarified both in the justification of the study and in its objective.
The methodology is excessively concise, it uses 38 items measuring 10 aspects of the culture of patient safety but does not explain the 10 survey domains in the measurements, nor the study period.
The statistical analysis should include the analysis method (Cronbach's) of the internal consistency of the tool, which the authors include in the result.
The results (table 2) show a comparison with the 2020 SOPS * MAedical Office Database (n = 19,396) without previously making any reference to this base and this acronym.
The results speak of classifications of general qualification in safety and general qualification in quality, when previously these concepts are not well clarified or defined
The discussion is very cumbersome, limiting itself to comparing the different domains with the results of other countries, without discussing the different groups of professionals or health systems in which they have been applied. It is recommended to lighten this discussion and reflect more on the results obtained. Nor do they analyze why they have such a low percentage, for example, of nursing professionals' response, ... nothing related to the pandemic is discussed either,
In the bibliography there are two repeated citations, 18 is the same as 23 and 21 is the same as 25.
The manuscript is interesting, the authors are encouraged to review it and they are congratulated for the work done
Reviewer 2 Report
Generally, the paper could use an edit. There are several awkward uses of phrasing. In the abstract the authors note that patient safety and quality of service is an urgent need during the pandemic. Patient safety and quality of service is an urgent need in general whether or not there is a pandemic. The paper presents no basis concluding that the pandemic has had an impact on patient safety and quality. Introduction: the background literature discussion is fairly terse and focuses on adverse events. The instrument used here does not relate to adverse events but rather to assessment of a culture of patient care using an instrument. Statistical analysis. The authors use SPSS to conduct independent sample t-tests, correlations using Pearson’s r. There is a later reference in the paper to a regression analysis that is not described in the statistical analysis section nor is the result reported. The authors have not evaluated the distribution of the variables. The independent samples t-test and the Pearson correlation both assumed normality of distribution. Since this is unknown for these data it would be more appropriate to use Mann-Whitney for investigating differences in means and the Spearman correlation coefficient for correlation. The authors compute a reliability statistic that they describe as Cronbach’s alpha. The developer of the test prefers the term “raw coefficient alpha” rather than Cronbach alpha. A description of this test should be included in the statistical analysis paragraph. Again, the authors should justify use of this measure of reliability as opposed to Spearman-Brown. In the results section the authors compare the average percent response in the current study with a 2020 SOPS Medical Office Database. There is no discussion of the Database comparison. This should be included in the introduction. Table 5 is labeled "correlations between respondents gender, job seniority, category of clinical staff and present positive scores on patient safety culture." However, this appears to be a difference in means test rather than a correlation. Moreover, there are almost no statistically significant results here. The authors might consider eliminating table 5 and replacing it with a shorter description of the statistically significant results. I am not convinced that these comparisons are particularly meaningful. Why are they important The top of page 6 discusses the linear regression model. These results are not reported. The authors should discuss their use of linear regression in the statistical analysis section and should report the model results here including the results of the F test, the adjusted R square and the model coefficients and P values. I can’t tell from this whether these linear regressions are single variable models or multivariate. If they are single variable models there are deep problems here. The Discussion section compares questionnaire results for this study with studies in other countries. The discussion is a run-on analysis. The single paragraph discussion goes on for 2 ½ pages. The authors should look for a way to present these results in a table and then to discuss some of the more important findings. Given that the study is the first to validate the Greek MSOPSC it would be or publication if the other issues discussed above can be dealt with.